# An Integrated Analysis of microRNAs and the Transcriptome Reveals the Molecular Mechanisms Underlying the Regulation of Leaf Development in Xinyang Maojian Green Tea (*Camellia sinensis*)

**DOI:** 10.3390/plants12213665

**Published:** 2023-10-24

**Authors:** Xianyou Wang, Ruijin Zhou, Shanshan Zhao, Shengyang Niu

**Affiliations:** 1School of Horticulture Landscape Architecture, Henan Institute of Science and Technology, Xinxiang 453003, China; persimmonzhou@163.com; 2Henan Province Engineering Research Center of Horticultural Plant Resource Utilization and Germplasm Enhancement, Xinxiang 453003, China; 3School of Food Science, Henan Institute of Science and Technology, Xinxiang 453003, China; zhaoshanshanzsh@163.com (S.Z.); niushengyang@163.com (S.N.)

**Keywords:** tea plant, miRNA, Xinyang Maojian, leaf development, regulatory network

## Abstract

Xinyang Maojian (XYMJ) tea is one of the world’s most popular green teas; the development of new sprouts directly affects the yield and quality of tea products, especially for XYMJ, which has hairy tips. Here, we used transcriptome and small RNA sequencing to identify mRNAs and miRNAs, respectively, involved in regulating leaf development in different plant tissues (bud, leaf, and stem). We identified a total of 381 conserved miRNAs. Given that no genomic data for XYMJ green tea are available, we compared the sequencing data for XYMJ green tea with genomic data from a closely related species (Tieguanyin) and the *Camellia sinensis* var. *sinensis* database; we identified a total of 506 and 485 novel miRNAs, respectively. We also identified 11 sequence-identical novel miRNAs in the tissues of XYMJ tea plants. Correlation analyses revealed 97 miRNA–mRNA pairs involved in leaf growth and development; the csn-miR319-2/csnTCP2 and miR159–csnMYB modules were found to be involved in leaf development in XYMJ green tea. Quantitative real-time PCR was used to validate the expression levels of the miRNAs and mRNAs. The miRNAs and target genes identified in this study might shed new light on the molecular mechanisms underlying the regulation of leaf development in tea plants.

## 1. Introduction

Tea (*Camellia sinensis*) is the world’s oldest and most popular nonalcoholic beverage; it also provides various health benefits [1]. These health benefits are conferred by several compounds, including tea polyphenols, catechin acid, caffeine, theanine, proteins, and alkaloids [2,3]. According to data from the China Tea Marketing Association in 2021 (http://www.stats.gov.cn/, 1 July 2023), approximately 11.7 million tons of fresh tea leaves are processed into green tea, black tea, white tea, and oolong tea annually.

Tea (Theaceae: *Camellia*) was first grown and produced in Yunnan Province, P.R. China for approximately 1700 years [4]. There are two main cultivated tea tree varieties: *Camellia sinensis* var. *sinensis* (CSS; Chinese type) and *Camellia sinensis* var. *assamica* (CSA; Assam type). CSS is one of the most popular tea varieties in China; it also has one of the most extensive distributions of any tea cultivar worldwide [4,5]. In China, CSS comprises approximately 67% of all tea cultivars, and CSS has been the source of germplasm that has contributed to recent increases in tea production. Xinyang Maojian (XYMJ) green tea is one of the most popular CSS teas in China. XYMJ was first grown in southern Henan Province, China, and later exported to more than 20 countries and regions. 

Tea plants are perennial evergreen woody plants that are grown in tropical and subtropical regions worldwide [6]. Tea plants have a lifespan of over 100 years; under natural conditions, they can reach heights of approximately 15 m. Tea is derived from one bud and the two uppermost leaves on the tender shoots. Given that young leaves from tea trees are used for tea preparation, the developmental characteristics of tea leaves are thought to have a major effect on the yield and quality of tea products [7]. 

The size of the tea genome is approximately 3.0 Gb [3,8]. It shows high heterozygosity, which is at least in part associated with genetic barriers (i.e., self-incompatibility), and this has greatly inhibited research on tea plant genomes. A total of 30 tea plant genomes have been published to date according to the tea plant genome database (TeaPGDB) (http://eplant.njau.edu.cn/tea, 1 July 2023). These include the whole genomes of six cultivars (CSA ‘Yunkang 10’, CSS ‘Biyun’, CSS ‘Shuchazao’, wild tea plant, CSS ‘Huangdan’, and CSS ‘Shuchazao’), as well as seven nuclear genomes and seventeen organellar genomes [9]. No genomic data are currently available for XYMJ green tea. 

MicroRNAs (miRNAs) are endogenous 20–24-nucleotide (nt) noncoding small RNAs (sRNAs) that play various regulatory roles by mediating the cleavage of their target mRNAs or repressing their translation in animals and plants [10,11,12]. In plants, miRNAs play key roles in diverse biological processes, such as growth and development, the degradation of proteins, signal transduction, responses to biotic and abiotic stress, and regulation of the expression of other sRNAs [13,14,15]. Leaf development is controlled by a complex gene regulatory network, and miRNAs and their target genes play key roles in regulating various aspects of this process [16,17,18]. Plant miRNAs and their target genes affect the external characteristics of leaves and determine leaf polarity [19,20]. Some novel miRNAs that play key roles in regulating the development of leaves in plants have been identified and cloned. For example, miR164 regulates leaf margin serration in Arabidopsis [21]; miR390 and miR165/166 control polar growth in Arabidopsis [19,20]; miR159, miR319, and miR396 control leaf shape formation [16,22]; and miR156, miR160, and miR164 play key roles in the initiation of leaf development [21,23,24]. The amount of miRNA information in tea plant genomes is increasing, and this has been driven by the development of high-throughput sequencing technology, which has become an important tool for the identification of miRNAs. The first four candidate tea miRNAs in a tea plant genome were identified using expressed sequence tag (EST)-based comparative genomics [25,26]. Since then, comparisons of tea EST sequences, transcriptome data, the Arabidopsis genome, and the tea genome have been used for miRNA identification. Several studies have examined the mechanisms underlying the ability of miRNAs to alter responses to abiotic and biotic stress through their effects on target genes; miRNAs have also been shown to regulate secondary metabolite biosynthesis in tea plants [27,28]. 

The leaves of tea plants are vegetative organs and economically valuable. Leaf size, shape, and color are some of the most important characteristics of tea plants, and these morphological features are affected by various factors such as light, mineral nutrients, hormones, temperature, and the expression of target genes regulated by miRNA. In China, one bud and two leaves of tea plants are harvested in the spring; these materials are commonly used to make green tea and are the most economically valuable components of tea plants. Several previous studies have shown that miRNAs play key roles in regulating the development of sprouts in several tea varieties, such as O. Kuntze var. Chinary tea [29], Shuchazao tea [30], Pingyang Tezaocha tea [7], and Yunkang 10 tea [31]. Some miRNA families that regulate the expression of plant-specific transcription factors are involved in multiple developmental processes. The miR319/TCP (TEOSINTE BRANCHED1, CYCLOIDEA, and PROLIFERATING CELL NUCLEAR ANTIGEN BINDING FACTOR, TCP) module is involved in multiple developmental processes in tea plants; for example, miR319c/TCP2 regulates apical bud burst in Shuchazao tea [32]. However, the molecular mechanisms and regulatory pathways underlying miRNA-mediated regulation in XYMJ tea plants remain unclear.

Here, we identified miRNAs by comparing XYMJ sequencing data against the Tieguanyin genome, CSS database, and Arabidopsis genomic database. Generally, our approach could be used to identify miRNAs in any nonreference genome sequence. The aims of this study were to identify the main candidate miRNAs and mRNAs involved in regulating leaf development and clarify the molecular mechanisms by which the miRNA–mRNA regulatory network controls leaf development.

## 2. Materials and Methods

### 2.1. Plant Materials 

We used three-year-old tea plants (*Camellia sinensis* var. ‘XYMJ’) in this study. The tea plants were planted in tea gardens from Yipin Tea Plantation in Xinyang, Henan Province, China, on 10 May 2022. Samples were collected from a total of 100 XYMJ tea plants. Buds (0.5 g) were harvested from XYMJ tea plants, and three biological replicate bud samples were collected. First leaf, second leaf, and stem samples were collected in the same manner. All samples of each sample type were collected using the same procedures to ensure that samples were standardized. Buds collected from the top of tea plants were 0 to 3 cm in length (sBud); the first leaves below the buds (sL1) were approximately 3 cm long; the second leaves with higher maturity below the buds (sL2) were approximately 4 cm long, and the stem between the first leaf and second leaf (sS1) was approximately 2 cm long. After samples were collected, they were immediately frozen in liquid nitrogen and maintained at −80 °C until RNA extraction.

### 2.2. Total RNA Extraction

The Quick RNA Isolation Kit (Waryong, RNAzol, Beijing, China) was used to extract total RNA per the manufacturer’s instructions. A QuantusTM Fluorometer (Promega Corporation, Madison, WI, USA) and 1% agarose gel electrophoresis were used to evaluate the purity and quantity of total RNA. Transcriptome and miRNA sequencing were then conducted using the RNA samples. 

### 2.3. sRNA and RNA Sequencing (RNA-Seq) Library Construction and Sequencing

Samples from four different tissues (bud, first leaf, second leaf, and stem) were used to generate four sRNA and RNA-seq libraries, and these libraries were sequenced by Personal Biotechnology Co. (Nanjing, China, http://www.personalbio.cn/, 1 July 2023) using the Illumina Novaseq 6000 platform. First, sRNA fragments (16–30 nt) were isolated after separation by a 15% polyacrylamide denaturing gel. The Small RNA Purification Kit (Waryong Centrifugal column type, Beijing, China) was used to purify the fragments. T4 RNA ligase was then used to sequentially ligate the purified products to a 5′ RNA adaptor and a 3′ RNA adaptor by T4 RNA ligase; they were then reverse-transcribed to generate double-stranded cDNA. An Agilent 2100 Bioanalyzer and Agilent High Sensitivity DNA Kit (Agilent Technologies Inc., Santa Clara, CA, USA) were used to examine the validated and purified sRNA-derived cDNA library. We then sequenced each sRNA library. All sRNA fragments were annotated, and the target genes of the novel miRNAs were predicted. The procedure used to construct RNA-seq libraries is based on that described in a previous study [33]. 

### 2.4. Identification of Conserved and Novel miRNAs

Potential conserved and novel miRNAs of *C. sinensis* were predicted via bioinformatics analysis of the sequencing data. Given that genomic data for XYMJ green tea are not available, HISAT2 (v.2.0.5) with default parameters was used to compare the sequencing data against transcriptome data and genomic data from *C. sinensis* Tieguanyin (http://eplant.njau.edu.cn/tea/download.html, 1 July 2023) [34], the CSS genome database (http://tpdb.shengxin.ren/index.html, 1 July 2023), and Arabidopsis genomic data (https://www.arabidopsis.org/download/index-auto.jsp?dir=%2Fdownload_files%2FGenes%2FAraport11_genome_release, 1 July 2023). Several RNA categories, such as repeat sequences, tRNAs, rRNAs, protein-coding genes, small nuclear RNAs (snRNAs), and small nucleolar RNAs (snoRNAs), were removed from each database by comparing the mapped sRNA tags against the Tieguanyin genome and CSS genome database. Following this analysis, known miRNAs that were mapped to the database sequences were categorized into different miRNA families. Novel miRNA candidates were classified as novel putative miRNAs and placed in a single group. The Mireap online program (http://sourceforge.net/projects/mireap/, 1 July 2023) was used to predict the novel miRNAs.

### 2.5. Annotation and Cluster Analysis of miRNAs

Unique reads were annotated using known miRNAs in the miRBase database (http://www.mirbase.org/, 1 July 2023) and other non-coding RNAs (ncRNAs). No annotated sequence information was obtained for the novel miRNAs that were identified using Mireap (version 2.0) software. To identify sRNAs derived from miRNA fragments, we compared the sense and antisense sequences of sRNAs and miRNAs using Overlap software. rRNA, snoRNA, snRNA, and tRNA sequences from the GenBank and Rfam3 databases were used to annotate sRNA sequences. rRNAs were given highest priority in the sequence comparisons, followed by conserved miRNAs, repeats, exons, and introns, to ensure that sRNA annotations were unique [35].

These miRNA sequences were aligned to mature miRNA sequences of closely related species in the miRBase database using BLAST to identify miRNAs that were evolutionarily conserved across species. The target genes for differentially expressed miRNA (DEM) sequences were predicted using psRobot_tar software [36]. Comparisons of the XYMJ sequencing data against different genome databases were conducted to identify DEMs in various libraries. DESeq (v1.30.0) was used to analyze the DEMs, which were identified using the following criteria: |log_2_(fold change)| > 1 and *p*-value < 0.05. Next, we used the pheatmap package in R software to conduct bidirectional cluster analysis of all samples. Euclidean distances were calculated, and samples were clustered using the hierarchical clustering method. 

### 2.6. Gene Ontology (GO) and Kyoto Encyclopedia of Genes and Genomes (KEGG) Analysis

GO (http://www.geneontology.org/, 1 July 2023) and KEGG (http://www.kegg.jp/, 1 July 2023) analyses were conducted on the target genes of the DEMs. GO analysis of the predicted target transcripts of miRNAs was conducted to clarify their regulatory roles in tea plants. The topGO package was used to conduct GO analysis on the target genes of DEMs; the hypergeometric distribution method was used to calculate *p*-values, and the threshold for significant enrichment was *p* < 0.05. GO terms with significantly enriched differentially expressed genes (DEGs) were identified to characterize the main biological functions performed by these genes. KEGG analysis of the target genes of DEMs was conducted using clusterProfiler (3.4.4) software, and the threshold for statistical significance was *p* < 0.05.

### 2.7. qRT-PCR Validation of miRNAs

qRT-PCR was conducted to validate the expression of miRNAs and their target genes. The total RNA from various tissues of XYMJ tea plants (bud, first leaf, second leaf, and stem tissue) used to construct the RNA-seq libraries was also used in qRT-PCR reactions. DNAman software (version 7.0; Lynnon Biosoft, Quebec, QC, Canada) was used to design the primers for mRNA and miRNA qRT-PCR assays, and the primer sequences are shown in Appendix A. The SYBR One Step PrimeScript miRNA cDNA Synthesis kit (Takara, Otsu, Japan) was used to synthesize the cDNA of the miRNAs following the manufacturer’s instructions. All qRT-PCR reactions were conducted on a Bio-Rad IQ5 instrument (Foster City, CA, USA) with the following thermal cycling conditions: 94 °C for 2 min, followed by 45 cycles of 94 °C for 15 s, 58–62 °C for 15 s, and 72 °C for 30 s. The 2 (-deltadeltaCT) method was used to determine the expression levels of the target genes of the miRNAs by calculating the fold change in expression in selected tissues relative to bud tissue. The expression of the *glyceraldehyde-3-phosphate dehydrogenase* (*GAPDH*), *5S-rRNA*, and *U6 small nucleolar RNA* genes were determined and used as reference genes [37]. Three biological replicates were conducted in all qRT-PCR analyses, and three technical replicates were performed for each biological replicate.

### 2.8. Target Prediction and Construction of a Regulatory and Interaction Network

The fold changes in expression (|log_2_(fold change)| > 1) and the significance of expression differences (*p*-value < 0.05) of DEMs were evaluated using DESeq. The target genes for the DEM sequences were predicted using psRobot_tar software [36]. To construct the network of miRNAs and mRNAs involved in regulating leaf development in XYMJ tea, we conducted searches of miRNAs and mRNAs against the Retrieval of Interacting Genes (STRING) database using the Search Tool. We further analyzed targeting relationships between DEMs and selected mRNAs and DEMs. GO and KEGG enrichment analyses were conducted to shed light on the roles of these miRNAs in leaf development. According to the protein interaction relationships in the STRING database, we selected interaction pairs with scores greater than 0.95 to construct the network map. We used Cytoscape software to alter the color and shape of the nodes in the network diagram. To further clarify interactions between mRNAs and miRNAs, differentially expressed miRNA families and mRNAs in the diagram were indicated using Photoshop software. 

### 2.9. Statistical Analysis

SPSS software (SPSS, Inc., Chicago, IL, USA) was used to conduct statistical analyses. One-way analysis of variance followed by Tukey’s post-hoc test was used to evaluate the significance of differences between groups; the threshold for statistical significance was *p* < 0.05. RT-qPCR results are shown as the mean ± SD of three independent experiments. Analysis of variance (ANOVA) was performed using Duncan’s multiple range test; different lowercase letters indicate significant differences at *p* < 0.05. 

## 3. Results

### 3.1. High-Throughput Sequencing of sRNAs in XYMJ Tea Plants 

Four sRNA libraries derived from various tissues and leaf development stages (sBud, sL1, sL2, and sS1) were sequenced using the Illumina platform to identify miRNAs involved in leaf development in XYMJ tea plants. Overlap software was used to compare all unique reads against the Rfam13 database. We identified a total of four types of known ncRNAs: rRNAs, tRNAs, snRNAs, and snoRNAs. Only RNAs that were perfect matches or with no more than two sequence mismatches were retained. A summary of the sequencing data of the four sRNA libraries is provided in Table 1. Clean reads were obtained by removing low-quality reads, reads with adaptor/acceptor sequences, as well as contaminants, including adaptor sequences and RNAs less than 20 nt. A total of 3,896,052 known ncRNAs were obtained from the four samples. The most and least abundant sRNAs in the library were rRNAs (83.4%) and snRNAs (2.5%), respectively (Table 1). 

### 3.2. Conserved miRNAs in XYMJ Tea Plants

Conserved miRNAs in XYMJ tea were identified by aligning XYMJ tea sRNAs that were not annotated as rRNA, tRNA, snRNA, and snoRNA to mature miRNAs and precursor miRNAs in all plant species uploaded to the miRBase 22.1 database (http://www.mirbase.org/, 1 July 2023). Only RNAs that were perfect matches or with no more than two sequence mismatches were retained. A total of 15,502 unique sequences and 675 conserved miRNAs from various tissues were obtained in XYMJ tea. A total of 167, 86, 189, and 233 conserved miRNAs were identified in the sBud, sL1, sS1, and sL2 samples, respectively (Appendix A).

The transcriptome data and Tieguanyin genome, CSS genome, and Arabidopsis genome databases were compared to characterize the extent to which these miRNAs are evolutionarily conserved. Comparison with the transcriptome data yielded 381 conserved miRNAs, and target genes were predicted for 107 conserved miRNAs. The number of target genes and target gene loci detected were 703 and 1116, respectively (Table 2 and Appendix A). The same numbers of conserved miRNAs were obtained by comparison of the Tieguanyin genome, CSS genome, and Arabidopsis genome databases. However, slight differences were observed in the number of target genes and target gene loci (Table 2). 

BLAST was used to compare the miRNAs identified against sequences in the Rfam13 database to classify conserved miRNA families. A total of 381 known conserved miRNAs were categorized into 60 families (Appendix A). The family with the largest number of miRNAs was the csn-miR166 family, which contained 39 members according to nucleotide differences. This was followed by csn-miR159 (36 members), csn-miR171 (28 members), csn-miR167 (22 members), csn-miR396 (18 members), csn-miR156 (17 members), csn-miR164 (14 members), csn-miR395 (13 members), and csn-miR172 (12 members). Only 1 to 10 members were present in the remaining miRNA families (Appendix A). We then used BLAST to align these miRNA sequences to mature miRNA sequences of closely related species in the miRBase database. Our findings indicate that the 381 known conserved miRNAs were highly conserved and belong to 47 known miRNA families (Appendix A). These findings suggest that the miRNA families in XYMJ tea plants are present in related plant species; their functions might be evolutionarily conserved in related plant species.

### 3.3. Characterization of Conserved miRNAs 

The frequency of different bases in miRNAs varied depending on base position and miRNA length. Our analysis revealed that adenine (A) was the first base in all 24 nt miRNAs, uracil (U) was the first base in most 20 to 23 nt miRNAs, and guanine (G) was the first base in most 23 nt miRNAs (Figure 1A). The percentages of A, U, G, and cytosine (C) at each locus of the conserved miRNAs in XYMJ tea plants were calculated. U is one of the most frequent bases in conserved miRNAs in XYMJ tea plants, and it was mainly observed at nucleotide positions 1, 2, 20, and 23. The distribution of A among positions was even. Although C was abundant in conserved miRNAs, it was less common at all positions, with the exception of positions 9, 19, 20, 21, 22, and 24 (Figure 1B).

### 3.4. Identification of Novel miRNAs in XYMJ Tea Plants

High-throughput sequencing analysis was conducted to identify miRNAs from the reference miRNA database, and novel miRNAs were identified using selected reference sequence information. Given that genome data for XYMJ green tea are not available, we compared the XYMJ sequencing data to genomic data from closely related species, including the Tieguanyin genome and CSS database. We obtained a total of 506 and 485 novel miRNAs (Table 2). Few novel miRNAs were obtained in Arabidopsis because of the differences between species. The results of the novel miRNA comparison varied depending on the species used (Table 2). 

Sequences of the novel miRNAs ranging from 20 to 24 nt in length are shown in Figure 2A. Most of the novel miRNAs were 24 nt in length, and sRNAs 24 nt in length were the most common. sRNAs 24 nt in length comprised 44.05% and 43.37% of the data obtained via comparison with the Tieguanyin and CSS genomes, respectively (Figure 2A). Sequences of known miRNAs ranged from 20 to 24 nt in length. The same numbers of miRNAs of different lengths were obtained by all four databases (Tieguanyin genome, CSS genome, Arabidopsis genome, and transcriptome). The length of most known miRNAs was 21 nt miRNAs, which comprised 62.73% of all miRNAs (Figure 2B). The characteristics of XYMJ tea plant miRNA sequences were consistent with the characteristics of miRNA sequences in *C. sinensis* observed in previous studies [7,30,38]. 

### 3.5. Tissue-Specific Expression of miRNAs

The tissue-specific expression of miRNAs might play a key role in regulating the growth and development of leaves. Thus, we determined the number of conserved and novel miRNAs showing tissue-specific and stage-specific expression in XYMJ tea plants across the four samples (sBud, sL1, sL2, and sS1). Some miRNAs were only expressed in a single tissue or stage. The greatest number of miRNAs were specifically expressed in buds (37 conserved miRNAs and 25 novel miRNAs). The fewest miRNAs were specifically expressed in stems (seven for conserved and seven for novel). A total of 20 miRNAs were specifically expressed in first leaves below buds, and 15 miRNAs were specifically expressed in second leaves below buds. The numbers of miRNAs specifically expressed in different tissues are shown in Appendix A.

### 3.6. Analysis of DEMs Involved in Leaf Development

Pairwise comparisons of DEMs (fold changes in expression greater than two and *p* ≤ 0.05) among sBud, sL1, sL2, and sS1 samples (Figure 3A) were conducted to identify miRNAs that play important roles in the development of XYMJ tea leaves. The numbers of DEMs between various tissues in XYMJ tea plants are summarized in Figure 3B,C. A total of 381 conserved miRNAs were obtained. Of these conserved miRNAs, 74 DEMs were identified in the sBud vs. sS1 comparison group (36 upregulated and 38 downregulated), and 19 DEMs were only detected in this comparison group. A total of 26 DEMs were identified in the sBud vs. sL1 comparison group (12 upregulated and 14 downregulated), and four DEMs were only detected in this comparison group. A total of 48 DEMs were identified in the sBud vs. sL2 comparison group (22 upregulated and 26 downregulated), and 15 DEMs were only detected in this comparison group (Figure 3B,C). We also identified 24 DEMs in the sBud vs. sL2 comparison group but only 3 DEMs in the sBud vs. sL1 comparison group (Appendix A). However, few DEMs were identified in the sL1 vs. sL2 comparison group (six upregulated and three downregulated) (Appendix A). Comparison of the XYMJ sequencing data against the Tieguanyin and CSS genome databases revealed 506 and 485 novel miRNAs, respectively (Table 2). We conducted a cluster analysis of the expression of all novel miRNAs to characterize the expression patterns of DEMs in different tissues of XYMJ tea plants. Known miRNAs were distinguished from novel miRNAs by the use of the two prefixes, ‘csn’ and ‘novel’, respectively, as has been performed in a previous study [30,32]. The suffixes ‘5p’ and ‘3p’ refer to the 5′ and 3′ arms of mature miRNA sequences. Cluster analysis of the data obtained from the comparison with each genomic database revealed nine clusters. The evolutionary relationships among each cluster of novel miRNAs were characterized. Single clusters contained novel miRNAs with the same expression profiles (Figure 3D,E).

Comparison of XYMJ unique reads to the Tieguanyin and CSS genome databases revealed 11 sequence-identical novel miRNAs in the tissues of XYMJ tea plants: novel-miR1952-5p, novel-miR1952-3p, novel-miR2171-3p, novel-miR2171-5p, novel-miR2361-5p, novel-miR2434-3p, novel-miR3308-5p, novel-miR3335-5p, novel-miR3562-5p, novel-miR3878-3p, and novel-miR4007-3p (Appendix A). The sequences of these 11 novel miRNAs were the same in the different genomic databases. The 11 novel miRNAs were present on the same chromosome. These findings suggest that the sequences on this chromosome might be highly conserved across tea species. We thus conducted further studies of these 11 novel miRNAs. 

### 3.7. DEG Analysis

We constructed transcriptomic sequence libraries to characterize the expression profiles of DEGs in different XYMJ tea plant tissues. DEGs were identified using the following criteria: *p*-value ≤ 0.05 and |log_2_ fold change| ≥ 2. A total of 265 DEGs were identified in different XYMJ tea plant tissues (Appendix A). We also conducted a trend analysis of differentially expressed mRNAs. The DEGs were divided into different clusters according to their expression patterns. A total of nine clusters were revealed by the trend analysis of the data (Appendix A).

### 3.8. Validation of Gene Expression by qRT-PCR

Several DEMs in different tissues of XYMJ tea plants were identified using high-throughput sequencing. To evaluate the accuracy of the high-throughput sequencing data and characterize the expression patterns of miRNAs, we conducted qRT-PCR analyses of eight conserved DEMs (csn-miR159-2, csn-miR159-45, csn-miR162-2, csn-miR166-22, csn-miR167-2, csn-miR168-6, csn-miR319-2, and csn-miR319-9), nine novel DEMs (novel-miR1952-5p, novel-miR1952-3p, novel-miR2171-3p, novel-miR2361-5p, novel-miR2434-3p, novel-miR2989-5p, novel-miR3335-5p, novel-miR3562-5p, and novel-miR4007-3p), and three predicted target genes (*csnTCP2*, *csnMYB33*, and *csnMYB104*) (Figure 4). 

The expression patterns of the miRNAs determined using high-throughput sequencing and qRT-PCR analyses were consistent. This indicates that the high-throughput sequencing data were robust and could be used to characterize the expression of miRNAs in the four libraries. According to the qRT-PCR analysis, the expression levels of three miRNAs (csn-miR167-2, csn-miR168-6, and novel-miR3562-5p) were higher in sBud than in sL1, sL2, and sS1. Four miRNAs (csn-miR159-2, csn-miR159-45, csn-miR319-2, and csn-miR319-9) were specifically expressed in sL1 and sL2. Three miRNAs (csn-miR162-2, csn-miR166-22, and novel-miR2171-3p) were specifically expressed in sS1, three miRNAs (csn-miR319-2, novel-miR1952-5p, and novel-miR3335-5p) were specifically expressed in sL1, and one miRNA (novel-miR4007-3p) were specifically expressed in sL2 (Figure 4). We used *U6* to confirm the patterns revealed by the miRNA expression analysis in different plant tissues. Our results indicated that the expression patterns of miRNAs for the two reference genes were consistent (Appendix A). Overall, our qRT-PCR data of the selected miRNAs indicated that the high-throughput sequencing data in the four libraries were reliable and accurate.

### 3.9. GO and KEGG Analyses 

GO and KEGG analyses were conducted to clarify the roles of DEMs in regulating leaf development in XYMJ tea plants. GO terms in the following three categories were assigned to DEMs to characterize their functions: molecular function, biological process, and cellular component (Figure 5A,C,E; Appendix A). A total of 1289, 305, and 189 GO terms were obtained in the biological process, molecular function, and cellular component categories, respectively. We analyzed the 10 most enriched GO terms. A total of 5339, 1325, and 1558 genes were enriched in GO terms in the biological process, cellular component, and molecular function categories, respectively. The greatest number of DEGs were enriched in GO terms in the biological process category (Appendix A). 

KEGG analysis was conducted to characterize associations between the targets of DEMs and specific biological processes; the 20 most enriched pathways were identified (Figure 5B,D,F; Appendix A). The three most enriched KEGG pathways were glycerolipid metabolism, fatty acid degradation, and phenylpropanoid biosynthesis for the sBud vs. sS1 comparison group; RNA polymerase, C5-branched dibasic acid metabolism, and ascorbate and aldarate metabolism were the most enriched KEGG pathways for the sBud vs. sL1 comparison group; and C5-branched dibasic acid metabolism, aminoacyl-tRNA biosynthesis, and MAPK signaling pathway were the most enriched pathways for the sBud vs. sL2 comparison group (Figure 5B,D,F). The most enriched pathway in the sBud vs. sL1 and sBud vs. sL2 comparison groups was plant hormone signal transduction, indicating that this process plays a key role in the development of the leaves of XYMJ tea plants (Figure 5D,F). Limonene and pinene degradation was also highly enriched in the sBud vs. sS1 and sBud vs. sL2 comparison groups (Figure 5B,F). In the sL1 vs. sL2 comparison group, the MAPK signaling pathway and amino acid metabolism were highly enriched. Furthermore, the caffeine metabolism pathways were enriched in the sL1 vs. sL2 comparison group (Appendix A). The enrichment of amino acid metabolism (including alanine, tyrosine, arginine, and proline metabolism) in all groups suggests that these pathways play key roles in the development of the leaves of XYMJ tea plants (Figure 5B,D,F). These findings suggest that specific miRNAs might play critically important roles in amino acid metabolism. 

### 3.10. Key DEMs Involved in Leaf Growth and Development

We analyzed miRNA expression to identify miRNAs that play a role in regulating the growth and development of the leaves. GO and KEGG analysis revealed 15 miRNAs that play a role in the development of the leaves of XYMJ tea plants. The expression patterns of 15 miRNAs, including 11 conserved and 4 novel miRNAs, as well as their target genes, are shown in Table 3. A total of 22 csn-miR167 family members and 7 csn-miR319 family numbers exhibited the same expression patterns in different tissues in XYMJ tea plants (Table 3 and Appendix A). 

### 3.11. Expression Analysis of miRNAs and Their Target mRNAs

The target genes of the DEMs were predicted to characterize the regulatory roles of miRNAs in the growth and development of the leaves of XYMJ tea plants. A total of 265 target DEGs were identified across all comparison groups (Appendix A). Next, we analyzed the predicted target DEGs of the DEMs to construct the regulatory network of miRNAs and mRNAs involved in the growth and development of the leaves. Correlation analyses were conducted to characterize the degree of involvement of the miRNAs in leaf growth and development. A total of 97 miRNA–mRNA pairs involved in the growth and development of the leaves were identified (Figure 6). Members of the csn-miR159, csn-miR319, csn-miR396, csn-miR167, and csn-miR171 families were the most highly connected conserved miRNAs. The relationships of the other conserved miRNAs in the network were relatively simple (Figure 6). 

## 4. Discussion

### 4.1. miRNAs Involved in the Development of the Leaves of XYMJ Tea Plants Were Expressed in Different Tissues and Developmental Stages

Processes related to the division, growth, and differentiation of cells that mediate the formation of complex tissues or organs such as leaves have been examined in several previous studies [39,40,41]. Leaves are derived from the shoot apical meristem, which is a group of stem cells located at the top of the shoot [42]. miRNAs are the main regulators of the expression of genes in plants, including leaf development, and they achieve their regulatory effects by mediating the cleavage of their target mRNAs or repressing their translation. Given that the leaves are the most economically valuable parts of tea plants, much effort has been made to enhance the growth and development of tea leaves. Here, we studied the functions of conserved miRNAs and compared XYMJ sequencing data against miRNA reference databases. We identified a total of 381 conserved miRNAs in XYMJ tea plants (Table 2). We also identified a few novel miRNAs by comparing the XYMJ sequencing data against the Arabidopsis genome database and XYMJ transcriptome. We compared the XYMJ sequencing data with the genomes of closely related species, including the Tieguanyin genome and CSS genome databases, and a total of 506 and 485 novel miRNAs were identified, respectively (Table 2). In addition, we compared the XYMJ sequencing data with the latest Arabidopsis genome database (TAIR11) (Table 2). More target genes of the miRNAs were obtained using the more complete version of the genome. These findings indicate that the conserved miRNA families present in XYMJ tea plants were also present in closely related plant species; this suggests that the functions of these miRNAs might be conserved in related plant species. Generally, a refined reference genome and annotation information can be used for the identification of miRNAs in species without reference genomes. Additional work is needed to identify novel miRNAs in XYMJ tea. 

Different miRNAs were identified in the sL1, sL2, and sBud samples, indicating that several different miRNAs play complementary roles in mediating the development of the leaves and buds. For example, both csn-miR167-2 and csn-miR168-6 were specifically expressed in the buds, and their expression levels were higher in sL1 than in sL2, suggesting that they might play a role in leaf elongation growth (Figure 4). The expression patterns of miRNAs in the different tissues of XYMJ tea plants varied. Differences in expression patterns were only observed between sS1 and sL1 in four miRNAs; differences in expression patterns were only observed between sS1 and sBud in 19 miRNAs; and differences in expression patterns were only observed between sS1 and sL2 in 16 miRNAs. The expression patterns of 43 miRNAs were the same across these three comparison groups (Figure 3B). We also identified 24 DEMs in the sBud vs. sL2 comparison group but only 3 DEMs in the sBud vs. sL1 comparison group (Appendix A). Few DEMs were identified in the sL1 vs. sL2 comparison group (six upregulated and three downregulated) (Appendix A). This might stem from the fact that sL1 and sL2 are more similar in external morphology. GO and KEGG analyses revealed that the MAPK signaling pathway and amino acid metabolism were highly enriched in the sL1 vs. sL2 comparison group. Caffeine metabolism pathways were enriched in the sL1 vs. sL2 comparison group (Appendix A). Single clusters revealed that the target genes of mRNAs in different samples had different expression profiles (Appendix A). This result is consistent with the miRNAs observed within a single variety of tea, especially in the first and second leaves. Other miRNAs were not included in these trends either because their expression patterns did not match the above patterns or because differences in their expression among developmental stages were not significant (*p*-value ≤ 0.05 and |log_2_ fold change| ≥ 2). The expression patterns observed in XYMJ tea plants were similar to those observed in ‘Pingyang Tezaocha’ tea [7]. 

Few studies have identified novel miRNAs and characterized their functions. To confirm the novel miRNAs identified in our study, we compared the unique reads of XYMJ tea plants with the Tieguanyin genome and CSS genome databases. A total of 11 identical novel miRNAs were identified (Appendix A). These 11 identical novel miRNAs exhibited different expression patterns in different tissues of XYMJ tea plants. Three novel miRNAs (novel-miR3308-5p, novel-miR3335-5p, and novel-miR3562-5p) were highly expressed in the sBud, sL1, and sS1 samples; two novel miRNAs were only expressed in one tissue (novel-miR2171-5p expressed in sL1 and novel-miR3878-3p expressed in sL2); and the expression patterns of the remaining novel miRNAs varied among the four samples (Figure 4, Appendix A). The sequences of these 11 novel miRNAs are the same in the Tieguanyin genome and CSS genome databases. The 11 novel miRNAs were clustered on the same chromosome. These findings indicate that the relevant sequence information on this chromosome might be highly conserved across different tea species. However, the regulatory mechanisms of these novel miRNAs in plants remain unclear.

In our study, the number of miRNAs that were specifically expressed in the buds was the highest among all miRNAs showing tissue-specific expression, and a total of 37 conserved miRNAs and 25 novel miRNAs with bud-specific expression were detected (Appendix A). This high percentage of miRNAs with bud-specific expression indicates that the number of miRNAs regulating mRNAs increases with development, especially in the bud (Figure 3, Appendix A). These expression patterns in XYMJ tea plants are consistent with those that have been documented in ‘Pingyang Tezaocha’ tea [7], ‘Yunkang 10’ tea [31], and ‘Fuding dabai’ tea [35]. 

### 4.2. Differentially Expressed Transcription Factors (TFs) Involved in the Development of the Leaves of XYMJ Tea Plants

Complex gene regulatory systems have evolved in higher plants to mediate various developmental processes, and miRNAs and TF are two well-studied sets of regulatory factors that play key roles at the transcriptional and posttranscriptional levels, respectively [43,44]. TFs are known to play important roles in the growth and development of the leaves of tea plants [45], secondary metabolism [8,46], and responses to biotic and abiotic stress [47,48]. For example, miR319 and the TCP gene family are involved in regulating leaf development [16]. TCP plays a key role in regulating plant growth, development, and senescence via the jasmonic acid synthesis pathway [49]. MYB TFs play important yet diverse roles in regulating leaf and stem development, secondary metabolite biosynthesis, and responses to environmental stress in tea plants [20]. In our study, a BLAST search of all the predicted target mRNAs against the PlantTFDB (http://planttfdb.cbi.pku.edu.cn/, version 4.0, 1 July 2023) was conducted to identify potential TFs. Several DEGs encoding members of TF families, including the MYB, bHLH, ERF, TCP, NAC, bZIP, and WRKY TF families, were identified based on the substantial differences in their expression among tissues in XYMJ tea plants (Appendix A). We identified a total of 579 MYB and 796 MYB-related genes (12.71%) that were differentially expressed in different tissues of XYMJ tea plants (Appendix A). The second largest group of TFs was the bHLH TFs (10.2%), followed by the ERF TFs (7.00%), NAC TFs (6.26%), WRKY TFs (4.20%), bZIP TFs (3.67%), and TCP TFs (2.53%). 

### 4.3. Network of miRNAs and mRNAs Involved in Regulating Leaf Development in XYMJ Tea Plants

miRNAs in plants regulate the expression of genes involved in tissue differentiation, embryonic development, and responses to environmental factors and stress [50]. Tea trees have been subjected to thousands of years of artificial selection, and at least 246 cultivars with major differences in morphology and physiology have been developed. XYMJ tea was first produced in the Dabie Mountains of Xinyang, Henan Province, China; it has thin, round, and straight leaves with a conspicuous gloss, hairy tip, and rich pekoe flavor. These are some of the characteristics used to distinguish among and breed new cultivars. Plant growth and development are accompanied by morphogenesis, and various regulators, such as sRNAs and TFs, might play important roles in both biological processes simultaneously. 

We analyzed the regulatory network of miRNA families and their target genes. A total of 97 miRNA–mRNA pairs were identified in the tissues of XYMJ tea plants. In these regulatory networks, members of the csn-miR159, csn-miR319, csn-miR396, csn-miR167, and csn-miR171 families were the most highly expressed (Figure 6, Appendix A). Generally, plant miRNAs might have diverse functions in complex regulatory networks; the main role of miRNAs is to suppress the expression of target genes [51]. Previous studies have suggested that the miR319 and *TCP* gene families are involved in regulating leaf development in Arabidopsis and Shuchazao tea [16,31]. In our study, the csn-miR319 family contained nine subgroups and 54 miRNAs. The csn-miR319-2 subgroup contained five miRNA members associated with five *TCP* genes (*TCP2*, *TCP3*, *TCP4*, *TCP10*, and *TCP24*) (Appendix A). Tissue-specific expression analysis revealed that the nine csn-miR319 subgroups contained 54 members with the same expression patterns in different tissues in XYMJ tea plants (Table 3 and Appendix A). We also verified the expression of csn-miR319-2 and its target gene *TCP2* using qRT-PCR. Our findings suggested that the expression of csn-miR319-2 was upregulated in sBud/sS1 and downregulated in sBud/sL1 and sBud/sL2. The expression of its target gene *TCP2* was downregulated in sBud/sS1 and upregulated in sBud/sL1 and sBud/sL2 (Figure 4). These findings indicate that *TCP2* might be a target gene of csn-miR319-2 and involved in the regulation of leaf development in XYMJ tea. The expression patterns of csn-miR319-2 and *TCP2* in XYMJ tea were not consistent with those previously observed in ‘Pingyang Tezaocha’ tea [7]. This might stem from differences in these varieties. These findings indicate that miR319 and its target *TCP* genes play key roles in leaf development. 

The functional roles of the miR159–MYB pathway in plant development have been studied in several plant species, such as Arabidopsis [51], rice [52], and wheat [53]. In our study, csn-miR159 was the largest family and contained the most target genes. The csn-miR159 family contained 43 miRNA members, which were associated with six *MYB* genes (Figure 6 and Appendix A). We verified the miRNA and mRNA expression patterns of the two groups of miR159–MYBs. The expression of csn-miR159-2 and csn-miR159-45 was upregulated in sBud/sS1 and downregulated in sBud/sL1, and sBud/sL2. The expression of their target genes *MYB33* and *MYB104* was downregulated in sBud/sS1 and upregulated in sBud/sL1 and sBud/sL2 (Figure 4, Appendix A). These findings were consistent with the high-throughput sequencing data. These results provide new insights into the molecular mechanisms underlying leaf development in XYMJ tea plants. Some novel miRNAs were identified in the tissues of XYMJ tea plants. The abundance of miRNAs, including novel miRNAs such as novel-miR3562-5p, novel-miR4007-3p, and novel-miR3335-5p, was higher in leaves than in the stem (Figure 4). These findings are consistent with the results of previous studies of ‘Pingyang Tezaocha’ tea [7]. In subsequent studies, we aim to explore the molecular mechanisms by which novel miRNAs regulate leaf development in XYMJ green tea.

## 5. Conclusions

We used RNA-seq and transcriptomic analyses to identify mRNAs and miRNAs in different plant tissues. We identified several conserved and novel miRNAs and their putative targets that might regulate the development of XYMJ leaves. GO and KEGG analyses indicated that specific miRNAs might play key roles in amino acid metabolism. Correlation analyses revealed 97 miRNA–mRNA pairs involved in the growth and development of the leaves in XYMJ green tea. The csn-miR319-2/csnTCP2 and miR159–csnMYB modules were involved in leaf development in XYMJ green tea. We plan to characterize the function of the csn-miR319-2/TCP2 and miR159–MYB modules in future studies. Clarification of the relationships between sRNAs can enhance the efficiency with which superior cultivars with higher quality could be bred.

## Figures and Tables

**Figure 1 plants-12-03665-f001:**
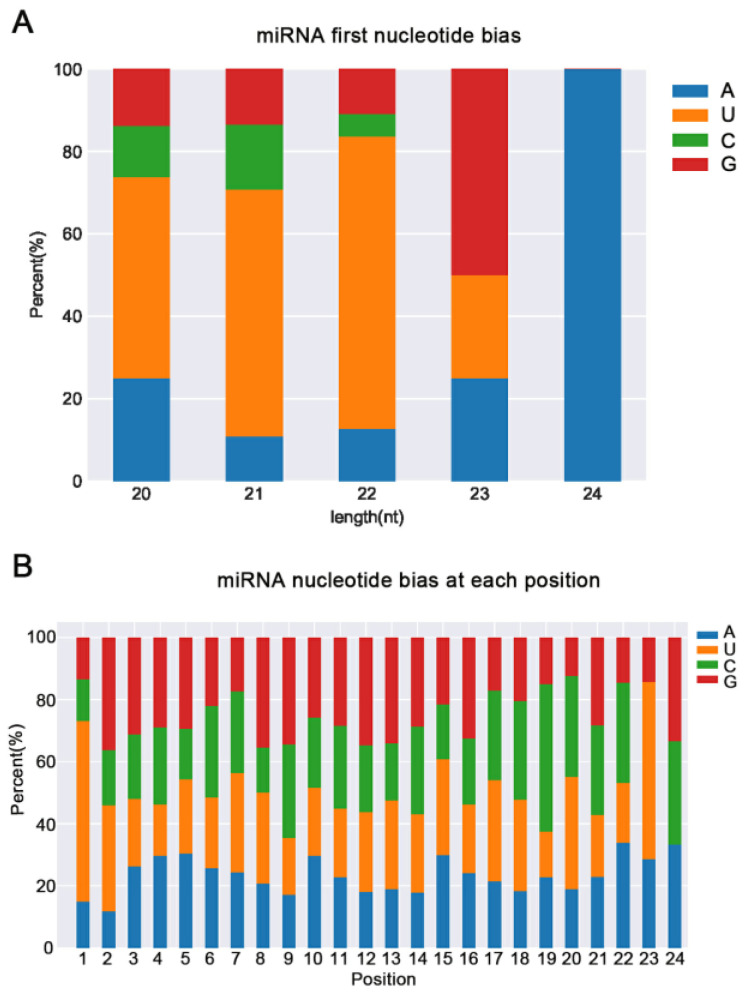
Nucleotide bias at all positions in conserved miRNAs in XYMJ tea plants. (**A**) The nucleotide bias of known miRNAs at the first position in XYMJ tea plants. (**B**) The frequency of A, U, C, and G of known miRNAs at each position. The X-axis shows miRNA nucleotide positions; the Y-axis shows the percentage of each nucleotide at each nucleotide position. Green, brown, blue, and pink correspond to A, U, C, and G, respectively.

**Figure 2 plants-12-03665-f002:**
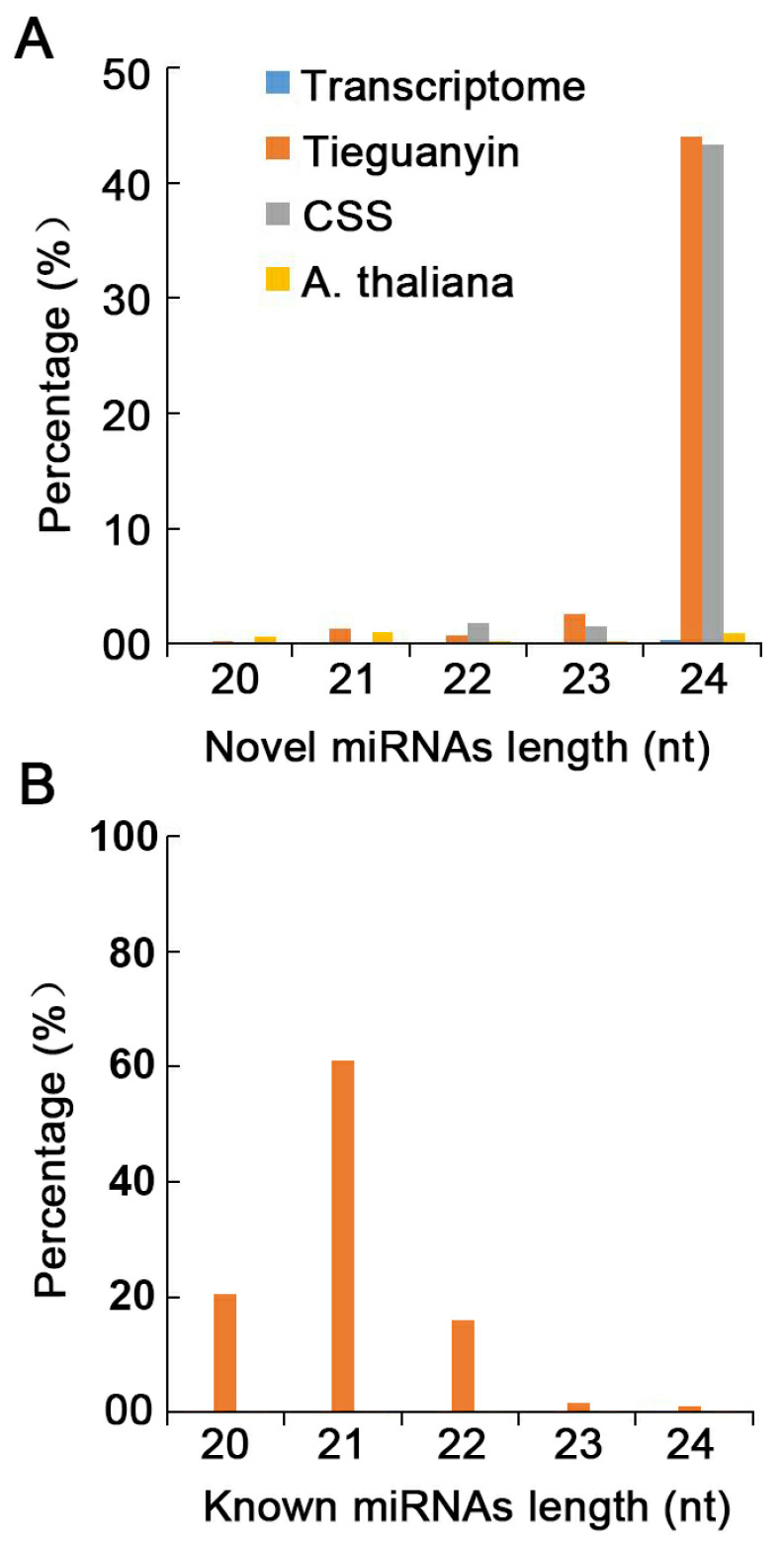
Distributions of miRNA of different sizes in XYMJ tea plants. (**A**) Distribution of novel miRNAs in four databases. (**B**) Distribution of known miRNAs of different lengths.

**Figure 3 plants-12-03665-f003:**
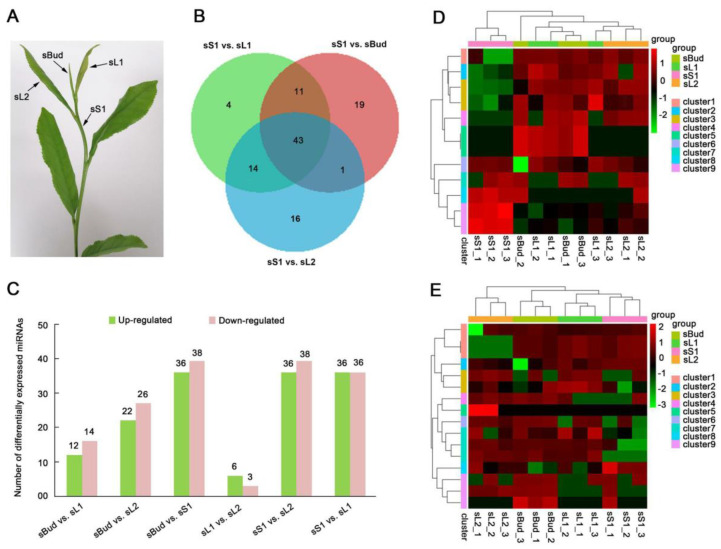
DEMs in different tissues. (**A**) Image showing the sBud, sL1, sL2, and sS1 samples that were collected. (**B**) Venn diagram showing the DEMs identified in each comparison group in XYMJ tea plants. (**C**) Number of upregulated and downregulated DEMs in each comparison group. Heat map of novel miRNAs in XYMJ tea plants showing differential expression among tissues compared with the CSS (**D**) and Tieguanyin (**E**) genome databases. Red indicates high miRNA expression, and green indicates low miRNA expression. miRNAs in the same cluster show similar expression patterns.

**Figure 4 plants-12-03665-f004:**
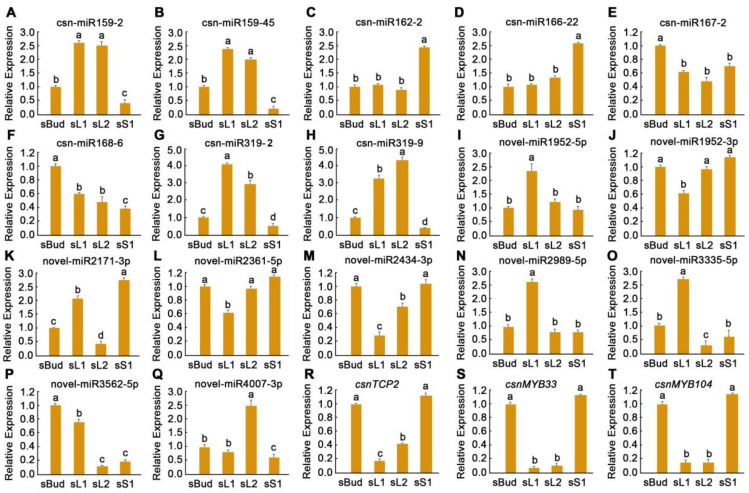
Relative expression of miRNAs and mRNAs in four tissues of XYMJ tea plants using qRT-PCR. (**A**–**H**) The expression levels of eight conserved miRNAs in different plant tissues. (**I**–**Q**) The expression levels of nine novel miRNAs in different plant tissues. (**R**–**T**) The expression levels of three target genes in different plant tissues. Error bars indicate the standard deviation (SD) of three independent biological replicates. Different letters above the bars indicate significant differences at *p* < 0.05. Means followed by the same letter over the bars are not significantly different at the 0.5% level.

**Figure 5 plants-12-03665-f005:**
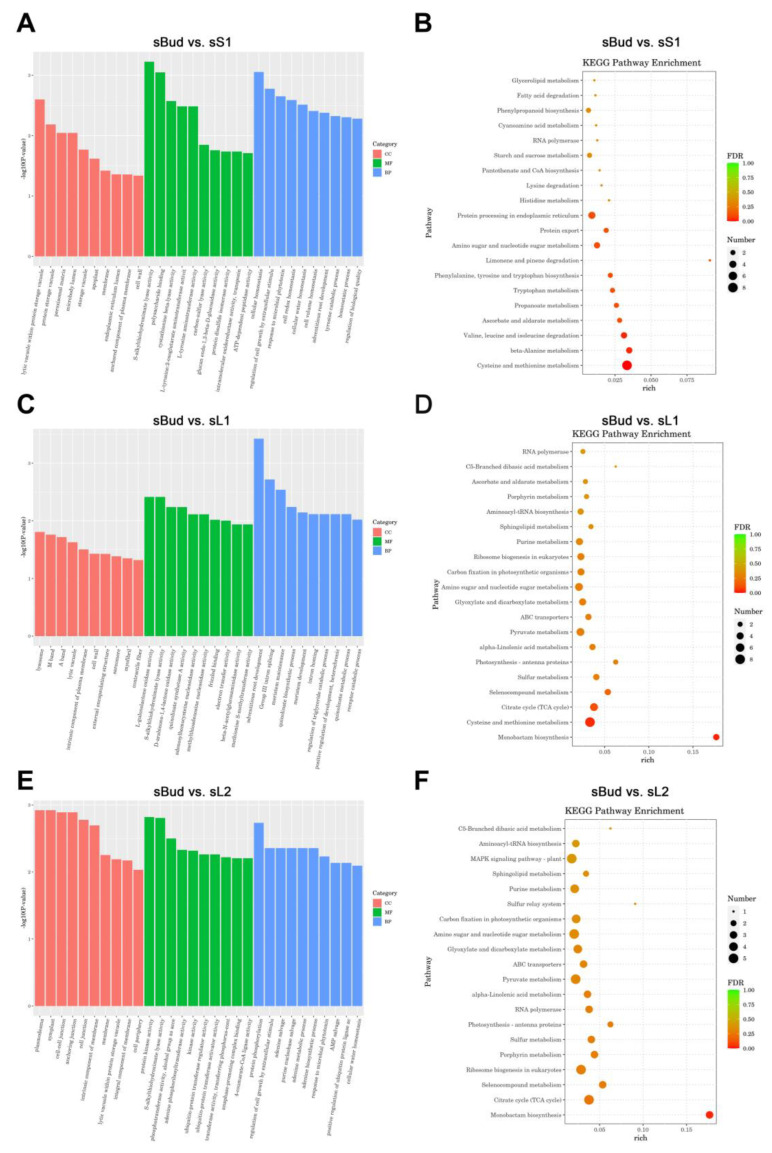
Results of GO and KEGG analyses of DEGs in different tissues of XYMJ tea plants. GO analysis was performed by assigning GO terms to DEGs in three categories: molecular function (MF), biological process (BP), and cellular component (CC). The 20 most enriched KEGG pathways of the target genes of the DEMs in XYMJ tea plants are shown. (**A**–**F**) show the GO terms and KEGG pathways for the sBud vs. sS1, sBud vs. sL1, and sBud vs. sL2 comparison groups. The significance of the matched gene ratio is indicated by the q-value. Redder values indicate higher q-values, and greener values indicate lower q-values; the size of the circle indicates the number of target genes.

**Figure 6 plants-12-03665-f006:**
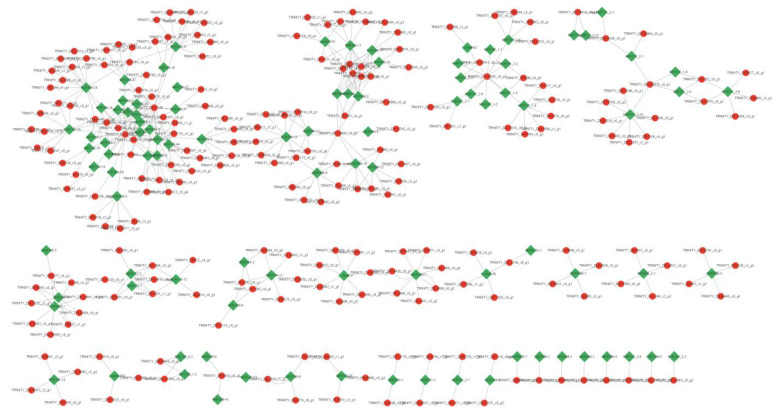
Analysis of the miRNA–mRNA regulatory network. Circles indicate potential target mRNAs, diamonds indicate miRNAs, and lines indicate miRNA–mRNA relationships. Red circles are upregulated genes, and green diamonds are downregulated genes.

**Table 1 plants-12-03665-t001:** Number of different sRNA types obtained in each sample.

Sample	rRNA	snoRNA	snRNA	tRNA
sBud-1	290,670	15,355	10,113	44,202
sBud-2	266,525	13,058	8667	38,303
sBud-3	223,086	13,483	9053	34,475
sL1-1	250,434	10,991	7016	33,634
sL1-2	300,386	10,672	7370	38,165
sL1-3	246,187	10,370	6873	31,690
sS1-1	227,361	10,842	7439	25,658
sS1-2	243,849	11,896	8133	28,680
sS1-3	252,037	12,271	8102	28,274
sL2-1	350,595	11,119	7909	38,158
sL2-2	298,649	11,791	8068	33,514
sL2-3	299,478	11,342	7768	32,341
Total	3,249,257	143,190	96,511	407,094

**Table 2 plants-12-03665-t002:** Summary of the results of the known and novel miRNA comparisons.

Comparison of Databases	Transcriptome	Tieguanyin	CSS	Arabidopsis
Known miRNA NO./novel miRNA NO.	381/4	381/506	381/485	381/32
Predicted target genes No. of miRNAs/novel miRNA NO.	107/0	103/2	104/7	99/3
No. of target genes in known miRNAs/novel miRNA	703/0	510/12	537/45	348/18
Predicted target genes locus No. of known miRNAs/novel miRNA	1116/0	861/12	864/56	1224/32

**Table 3 plants-12-03665-t003:** The miRNAs predicted to play key roles in the development of the leaves of XYMJ tea plants.

miRNA	Expression Pattern	Target Genes
csn-miR156-14	Downregulated (sBud/sS1, sBud/sL1, sBud/sL2)	*AT5G50590*, *AT5G50690*
csn-miR156-26	Upregulated (sBud/sS1, sBud/sL2)	*AT5G50960*, *AT3G14270*, *AT2G45820*
csn-miR159-2	Upregulated (sBud/sS1), downregulated (sBud/sL1, sBud/sL2)	*AT5G67090*, *AT3G06450*, *AT2G26950*
csn-miR159-45	Upregulated (sBud/sS1), downregulated (sBud/sL1, sBud/sL2)	*AT3G57060*, *AT2G03250*, *AT4G18390*, *AT2G13610*, *AT3G19930*, *AT2G43430*
csn-miR160-2	Downregulated (sBud/sS1, sBud/sL1, sBud/sL2),	*AT1G77850*, *AT2G28350*, *AT4G30080*
csn-miR162-2	downregulated (sBud/sS1)	*AT3G01330*, *AT2G23180*
csn-miR166-22	Downregulated (sBud/sS1, sBud/sL2)	*AT1G07810*, *AT1G30490*, *AT1G52150*
csn-miR167-2	Upregulated (sBud/sS1, sBud/sL1, sBud/sL2)	*AT5G41300*, *AT2G48110*, *AT3G57800*
csn-miR168-6	Upregulated (sBud/sS1)	*AT5G07140*, *AT5G17780*, *AT3G07195*
csn-miR319-2	Upregulated (sBud/sS1), downregulated (sBud/sL1, sBud/sL2)	*AT3G57090*, *AT3G61850*, *AT2G07689*
csn-miR319-9	Upregulated (sBud/sS1), downregulated (sBud/sL1, sBud/sL2)	*AT5G14820*, *AT5G46530*, *AT5G56970*, *AT3G30739*, *AT3G32980*, *AT3G62470*
novel-miR3335-5p	Downregulated (sBud/sS1)	
novel-miR3562-5p	Upregulated (sBud/sS1, sBud/sL1, sBud/sL2)	
novel-miR4007-3p	Downregulated (sBud/sL2)	
novel-miR4150-5p	Downregulated (sBud/sL1)	*CAS037071*, *CAS096907*

## Data Availability

The RNA-seq data of *C. sinensis* were deposited in the NCBI SRA database under the accession number PRJNA930208. The sRNA sequence data from this study were deposited in the Gene Expression Omnibus (GEO) database under the accession number GSE224544. The authors confirm that all the experimental data are available and accessible via the main text and the Appendix A.

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
