# Peer review of "An Integrated Analysis of microRNAs and the Transcriptome Reveals the Molecular Mechanisms Underlying the Regulation of Leaf Development in Xinyang Maojian Green Tea (Camellia sinensis)"

_plants, 2023, doi:10.3390/plants12213665_

Round 1

Reviewer 1 Report

The study "An Integrated Analysis of microRNAs and the Transcriptome
Reveals the Molecular Mechanisms Underlying the Regulation
of Leaf Development in Xinyang Maojian green tea (Camellia
sinensis)" sheds light on the molecular mechanisms underlying the regulation of leaf development in tea plants.

The study is interesting and the findings are important, I have one comment

Statistical significance is missing in all the figures

Moderate editing of English language required

Author Response

Reviewer 1

The study "An Integrated Analysis of microRNAs and the Transcriptome

Reveals the Molecular Mechanisms Underlying the Regulation of Leaf Development in Xinyang Maojian green tea (Camellia sinensis)" sheds light on the molecular mechanisms underlying the regulation of leaf development in tea plants. The study is interesting and the findings are important, I have one comment. Statistical significance is missing in all the figures. Moderate editing of English language required.

Answer: We thank the reviewer for pointing this out. We apologize for missing statistical significance in the figures. We added the difference significance analysis in figure 4 and figure S3 in the revised manuscript. Other figures are only quantitative statistics and do not require difference significance analysis.

The revised manuscript has been carefully reviewed by an experienced editor whose first language is English and who specializes in editing papers written by scientists whose native language is not English.

Reviewer 2 Report

Dear Author,

         I have read your manuscript. Since the study is very much needed in economical aspects, I have the following concerns,

        1. Author mentioned in the title that this manuscript underly the leaf development of tea. But it was mainly compared with buds and stem. So, the title and analysis were slightly different.

        2. In introduction, it was mentioned as buds and two leaves are harvested. Is there any significant correlation in the aspects of economical importance present in both buds and leaves. Did author find any clues in the metabolic pathways whether converged or diverged between these two tissues.

         3. As leaf is the most important tissues and its development is the main focus, there is less focus in the development of L1 and L2. I think author should give more focus on omics data between these tissue. Since both leaves are harvesting, what is the main differences between L1 and L2. what is author's conclusion on studying transcriptome and miRome from these tissues on particular.

      4. Under DEMs involved in the leaf development, bud and stem also compared. This dilute the main theme. You can either separately give the sud heading for other tissues or change the topic according to the results you are giving in the sub-section.

      5. I hardly see authors interpretation between two leaf tissues in discussion also.

All the best!

Reviewer 3 Report

Dear Authors, below are detailed comments on the manuscript:

1) add the purpose of the work to "Abstract",

2) "Camellia sinensis" - the form of writing should be standardized (I suggest writing it in italics),

3) 2.9. Statistical Analysi - the analysis of variance was probably in the parametric version, add information that the normality of the distribution of the data population and the homogeneity of variance in the samples were tested (what tests, what were their results and whether it allowed the use of parametric ANOVA),

4) Figures 1 and 2 - adapt to MDPI editorial requirements,

5) Figure no. 4 "...Data are mean ± SD..." - add this information to chapter 2.9.,

6) Figure 5. Results of .... - descriptions are illegible (should be corrected),

7) Figure 6. Analysis of the miRNA–mRNA ... as above,

8) measurement error should be marked on all bar charts (vertical error bars, also add information about this to chapter 2.9),

9) 5. Conclusions ... this chapter is very poor - it should be expanded

Author Response

Reviewer 3

Comments and Suggestions for Authors

Dear Authors, below are detailed comments on the manuscript:

1) add the purpose of the work to "Abstract",

Answer: We thank the reviewer for this comment. We have added. (Revised manuscript, page 1)

2) "Camellia sinensis" - the form of writing should be standardized (I suggest writing it in italics),

Answer: We thank the reviewer for pointing this out. We have revised it.

3) 2.9. Statistical Analysi - the analysis of variance was probably in the parametric version, add information that the normality of the distribution of the data population and the homogeneity of variance in the samples were tested (what tests, what were their results and whether it allowed the use of parametric ANOVA).

Answer: We thank the reviewer for pointing this out. We have added.

4) Figures 1 and 2 - adapt to MDPI editorial requirements,

Answer: We thank the reviewer for pointing this out. We have checked figures 1 and 2 to meet the MDPI editorial requirements.

5) Figure no. 4 "...Data are mean ± SD..." - add this information to chapter 2.9.,

Answer: Thank you for this valuable feedback. We have added.

6) Figure 5. Results of .... - descriptions are illegible (should be corrected),

Answer: We thank the reviewer for pointing this out. We have revised it.

7) Figure 6. Analysis of the miRNA–mRNA ... as above,

Answer: We thank the reviewer for pointing this out. We have revised it.

8) measurement error should be marked on all bar charts (vertical error bars, also add information about this to chapter 2.9),

Answer: We thank the reviewer for pointing this out. We apologize for missing statistical significance in the figures. We added the difference significance analysis in figure 4 and figure S3 in the revised manuscript. These information were added to the revised manuscript chapter 2.9.

9) 5. Conclusions ... this chapter is very poor - it should be expanded

Answer: We thank the reviewer for pointing this out. We have expanded the information in conclusion setion.

Round 2

Reviewer 3 Report

I would like to thank the Authors for introducing changes and additions